

# Sex differences in pacing during 'Ultraman Hawaii'

Beat Knechtle[1,2] and Pantelis T. Nikolaidis[3]

[1] Gesundheitszentrum St. Gallen, St. Gallen, Switzerland
[2] Institute of Primary Care, University of Zurich, Zurich, Switzerland
[3] Exercise Physiology Laboratory, Nikaia, Greece

## ABSTRACT

**Background**. To date, little is known for pacing in ultra-endurance athletes competing in a non-stop event and in a multi-stage event, and especially, about pacing in a multi-stage event with different disciplines during the stages. Therefore, the aim of the present study was to examine the effect of age, sex and calendar year on triathlon performance and variation of performance by events (i.e., swimming, cycling 1, cycling 2 and running) in 'Ultraman Hawaii' held between 1983 and 2015.

**Methods**. Within each sex, participants were grouped in quartiles (i.e., Q1, Q2, Q3 and Q4) with Q1 being the fastest (i.e., lowest overall time) and Q4 the slowest (i.e., highest overall time). To compare performance among events (i.e., swimming, cycling 1, cycling 2 and running), race time in each event was converted in $z$ score and this value was used for further analysis.

**Results**. A between-within subjects ANOVA showed a large sex $\times$ event ($p = 0.015$, $\eta^2 = 0.014$) and a medium performance group $\times$ event interaction ($p = 0.001$, $\eta^2 = 0.012$). No main effect of event on performance was observed ($p = 0.174$, $\eta^2 = 0.007$). With regard to the sex $\times$ event interaction, three female performance groups (i.e., Q2, Q3 and Q4) increased race time from swimming to cycling 1, whereas only one male performance group (Q4) revealed a similar trend. From cycling 1 to cycling 2, the two slower female groups (Q3 and Q4) and the slowest male group (Q4) increased raced time. In women, the fastest group decreased (i.e., improved) race time from swimming to cycling 1 and thereafter, maintained performance, whereas in men, the fastest group decreased race time till cycling 2 and increased it in the running.

**Conclusion**. In summary, women pace differently than men during 'Ultraman Hawaii' where the fastest women decreased performance on day 1 and could then maintain on day 2 and 3, whereas the fastest men worsened performance on day 1 and 2 but improved on day 3.

Corresponding author
Beat Knechtle,
beat.knechtle@hispeed.ch

# INTRODUCTION

'Ultraman Hawaii' is one of the longer distance triathlon lasting three days and consisting of 10 km swimming, 165 km cycling (day 1), 261 km cycling (day 2) and 85 km running (day 3). These distances are much longer than the typical Ironman triathlon events (3.8 km swimming, 180 km cycling and 42 km running) (*Seedhouse, Walsh & Blaber, 2006*). Performance in Ironman triathlon events is related to physiological measures, such as

maximal oxygen uptake, anaerobic threshold, economy of movement, hydration and energy homeostasis (*Laursen & Rhodes, 2001*). Another parameter that might play a role on the performance in 'Ultraman Hawaii' is pacing. Although pacing has been studied extensively in separate endurance and ultra-endurance events (*Knechtle et al., 2015a*; *Nikolaidis & Knechtle, 2016*) and in shorter formats of triathlon (*Wu et al., 2016*; *Wu et al., 2015*), no information on pacing in a multi-day triathlon such as 'Ultraman Hawaii' is available, i.e., how the performance in one event (e.g., swimming) influences the performance of subsequent events (e.g., cycling).

Pacing is defined as time per distance, usually expressed in minutes per kilometer or mile (*Andrew & Remco, 2012*). A pacing strategy—or a plan how to distribute an athlete's potential—is crucial for successful athletic practice (*Foster et al., 2005*) and has a considerable effect on performance in endurance sports (*Abbiss & Laursen, 2008*; *Foster et al., 1993*). *Abbiss & Laursen (2008)* postulated six different pacing strategies such as negative pacing (i.e., increase in speed over time), positive pacing (i.e., continuous slowing over time), all-out pacing (i.e., maximal speed possible), even pacing (i.e., same speed over time), parabolic-shaped pacing (i.e., positive and negative pacing in different segments of the race) and variable pacing (i.e., pacing with multiple fluctuations). *Abbiss & Laursen (2008)* stated that athletes in endurance sports often adopt a positive pacing strategy. However, they did not exclude the possibility that an even pacing strategy may be optimal to successfully complete an endurance event. These pacing strategies concern a single race, but their principles can be applied equally in a triathlon where performance might vary from an event to another event.

In a triathlon, the regulation of pacing may be influenced by different intrinsic and extrinsic factors (*Wu et al., 2014*). Moreover, the role of athletes' physiological state and emotional status has also been recognized (*Baron et al., 2011*). A complex system has been proposed to be involved in pacing regulation, dealing with elevated temperature, altered oxygen content of air, reduced fuel availability and subjective perception of distance covered (*Tucker & Noakes, 2009*). Most studies investigated pacing strategies for short distances such as the sprint distance triathlon (*Wu et al., 2016*), the Olympic distance triathlon (*Wu et al., 2015*), the Half-Ironman triathlon (*Wu et al., 2015*) and the Ironman triathlon (*Johnson et al., 2016*). Only a few studies investigated pacing strategies in very long-distance triathlons (*Knechtle et al., 2014*). However, no study investigated the pacing strategy in a three-stage ultra-triathlon such as 'Ultraman Hawaii.'

There is evidence that performance in split disciplines has an effect of overall race performance in a triathlon (*Le Meur et al., 2011*; *Vleck, Bürgi & Bentley, 2006*; *Wu et al., 2016*; *Wu et al., 2015*). In a sprint distance triathlon, faster cycling split times and overall race times were achieved with positive swimming pacing (*Wu et al., 2016*). For the Olympic distance triathlon, *Le Meur et al. (2011)* investigated the performance during the running split of 107 finishers in an Olympic distance triathlon and stated that elite triathletes should reduce their initial running speed for better performances during competitions. *Bernard et al. (2009)* investigatedten elite triathletes during the cycling split in an Olympic distance triathlon and showed that a progressive decrease in speed combined with an increase in speed

variability occurred during the race. A comparison between Olympic distance and Half-Ironman distance triathlon showed pacing strategies during a triathlon race are strongly influenced by both the distance and discipline (*Wu et al., 2015*). In the Ironman triathlon, pacing in downhill segments in the cycling and running split have an influence on overall race performance (*Johnson et al., 2016*).

There is also evidence that performance decreases during a multi-stage ultra-endurance triathlon (*Herbst et al., 2011*; *Knechtle et al., 2014*). In race held over ten times the Ironman distance, performance progressively declined over days (*Knechtle et al., 2015a*; *Knechtle et al., 2014*). Additionally, sex has an influence on pacing during triathlon (*Le Meur et al., 2009*; *Vleck et al., 2008*). In the Olympic distance triathlon, female and male elite triathletes adopted similar positive pacing strategies during swimming and running. Female triathletes were more affected by changes in slope during the cycling and running splits (*Le Meur et al., 2009*).

To date, little is known for pacing in ultra-endurance athletes competing in a non-stop event (*Heidenfelder et al., 2016*; *Knechtle et al., 2015b*) and in a multi-stage event (*Herbst et al., 2011*). In particular, we have no knowledge about pacing in a multi-stage event with different disciplines during the stages. Such knowledge would be of practical value for both coaches and athletes in order to optimize their pacing strategy according to sex and performance.

Therefore, the aim of the present study was to investigate pacing strategies in ultra-triathletes competing in 'Ultraman Hawaii'. This race is held since 1983 as a three-stage race with swimming and cycling on day 1, cycling on day 2 and running on day 3 (*Meili et al., 2013*). We hypothesized that athletes who would start fast on day 1 would slow down more compared to athletes who start slow on day 1. On the contrary, athletes who would start slow in the race would be able to improve during the race. Furthermore, we hypothesized to find differences in pacing between women and men.

## METHODS

To test our hypotheses, all women and men who finished 'Ultraman Hawaii' between 1983 and 2015 were considered. Sex, age and split times for the three stages were recorded and analyzed.The subjects for this study were obtained from official race website of 'Ultraman Hawaii' at http://ultramanworld.com. 'Ultraman Hawaii' is held as the Ultraman World Championship on the Big Island of Hawaii, USA. The race is limited to 40 athletes on an invitation-only basis and attracts participants from around the world. Racers must have reached their 20th birthday prior to the start of stage one. Each racer must be accompanied by an individual support team of at least two people over the entire course. The first edition of 'Ultraman Hawaii' was in 1983. Since then, the race was continuously held until today with exception of 1987 and 1991 where no race was organized. 'Ultraman Hawaii' is a three-day, 515 km (320 mile) race. The race is divided into three stages over three days: The first is a 10-km ocean swim from Kailua Bay to Keauhou Bay, followed by a 145-km cross-country bike ride, with vertical climbs that total ~1,800 m. Stage two is a 276-km bike ride from Volcanoes National Park to Kohala Village Inn in Hawi, with total vertical
climbs of ~1,200 m. Stage three is a 84-km double-marathon, which starts at Hawi and finishes on the beach at the Old Kona Airport State Recreation Area. Each stage must be completed within 12 h or faster. The swim portion of stage one must be completed in 5.5 h or faster. Participants who do not reach the finish lines within the time limits are disqualified. All procedures used in the study were approved by the Institutional Review Board of Kanton St. Gallen, Switzerland, with a waiver of the requirement for informed consent of the participants given the fact that the study involved the analysis of publicly available data (01/06/2010). Data from 1983 to 2015 were obtained from the official race website of 'Ultraman Hawaii' at http://ultramanworld.com. Age, sex, split times for the three stages (i.e., swimming and cycling on day 1, cycling on day 2 and running on day 3) were used for analysis.

## STATISTICAL ANALYSES

All statistical analyses were performed using IBM SPSS v.20.0 (SPSS, Chicago, USA). Descriptive statistics (mean ± standard deviations) were calculated for all data. Within each sex, participants were grouped in quartiles (Q1, Q2, Q3 and Q4) with Q1 being the fastest (i.e., lowest overall time) and Q4 the slowest (highest overall time). To compare performance among events (i.e., swimming, cycling 1, cycling 2 and running), time in each event was converted in $z$ score and this value was used for further analysis. Since race duration and race time variation differed among events, it was hard to compare race times of different events; thus, $z$ score was calculated from the formula score—mean/standard deviation and its use allowed comparing performance in different events. A between-within subjects analysis of variance (ANOVA) examined the effect of sex and overall performance (Q1, Q2, Q3 and Q4), as well as their interaction, on variation among events (i.e., swimming, cycling 1, cycling 2 and running). Sex and overall performance were defined as between subjects factors, and performance in events as within subjects factor. Bonferroni post-hoc analysis tested differences among groups. The magnitude of these differences was examined using eta square ($\eta^2$), classified as trivial ($\eta^2 < 0.01$), small ($0.01 \leq \eta^2 < 0.06$), medium ($0.06 \leq \eta^2 < 0.14$) and large ($\eta^2 \geq 0.14$). To study differences in performance and age by sex and calendar year from 1983 to 2015, we used a mixed-effects regression model with triathletes as random variable, whereas sex and calendar year were assigned as fixed variables. In addition, we examined the variation of age by calendar year using a mixed-effects regression model with triathletes as random variable, and sex and calendar year as a fixed variable. We examined interaction effects among these fixed variables. Akaike information criterion (AIC) was used to select the final model.

## RESULTS

Men did not differ from women for age (−0.6 years (−2.1; 1.0), mean difference (95% confidence interval), swimming time (−0:04:52 h:min:s (−0:13:32; 0:03:46)) and running (−0:15:23 h:min:s (−0:34:32; 0:03:46))) (Table 1). Men were faster in the overall race (−1:08:21 h:min:s (−1:51:36; −0:25:07)), cycling 1 (−0:18:06 h:min:s (−0:30:09; −0:06:03)) and cycling 2 (−0:29:59 (−0:45:14; −0:14:43)).

**Table 1 Performance time in participants by event.**

| | Total ($n = 765$) | Women ($n = 120$) | Men ($n = 645$) |
|---|---|---|---|
| Age (years) | $39.4 \pm 8.0$ | $39.9 \pm 8.4$ | $39.3 \pm 7.9$ |
| Overall race (h:min:s) | $29:28:50 \pm 3:42:48$ | $30:26:25 \pm 3:27:52$ | $29:18:06 \pm 3:44:00^{*}$ |
| Swimming (h:min:s) | $3:43:58 \pm 0:44:21$ | $3:48:05 \pm 0:43:02$ | $3:43:12 \pm 0:44:35$ |
| Cycling 1 (h:min:s) | $6:28:08 \pm 1:02:02$ | $6:43:24 \pm 1:04:35$ | $6:25:17 \pm 1:01:11^{*}$ |
| Cycling 2 (h:min:s) | $9:53:29 \pm 1:18:54$ | $10:18:46 \pm 1:22:06$ | $9:48:47 \pm 1:17:27^{\dagger}$ |
| Running (h:min:s) | $9:23:14 \pm 1:38:16$ | $9:36:12 \pm 1:31:31$ | $9:20:49 \pm 1:39:22$ |

**Notes.**
$^{*}p < 0.01$
$^{\dagger}p < 0.001$

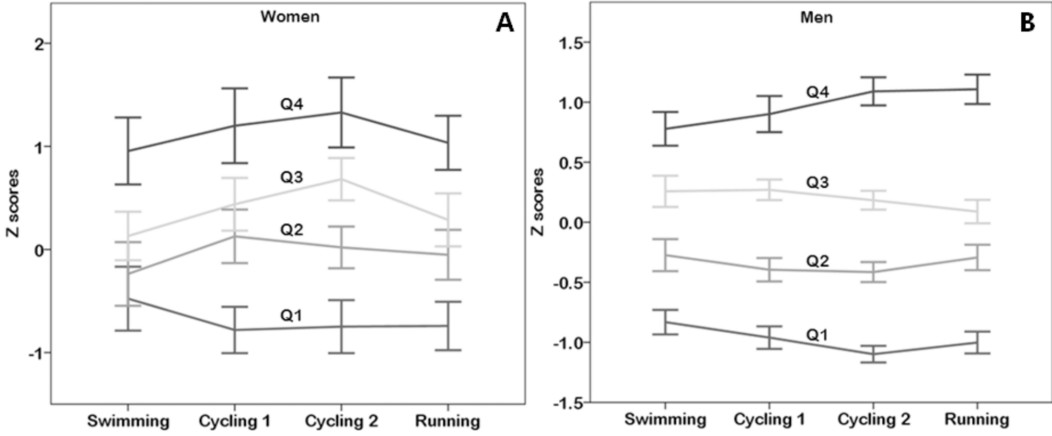

**Figure 1  Variation of performance in events by overall performance level and sex.**

Different trends among events were observed among groups differing for overall performance and between sexes (Fig. 1). The between-within subjects ANOVA showed a sex × event ($p = 0.015, \eta^2 = 0.014$, small magnitude) and a performance group × event interaction ($p = 0.001, \eta^2 = 0.012$, small magnitude). No main effect of event on performance was observed ($p = 0.174, \eta^2 = 0.007$, trivial magnitude). With regard to the sex × event interaction, three female performance groups (Q2, Q3 and Q4) increased race time from swimming to cycling 1, whereas only one male performance group (Q4) revealed a similar trend. From cycling 1 to cycling 2, the two slower female groups (Q3 and Q4) and the slowest male group (Q4) increased race time.

In women, the fastest group (Q1) decreased race time from swimming to cycling 1 and thereafter, maintained performance, whereas in men, the fastest group (Q1) decreased race time till cycling 2 and increased it in the running.

With regard to the participation of women and men across calendar years, there was an association between sex and calendar years, which was that women and men did not show a similar participation: an increased participation of women was observed during the last years (Fig. 2).

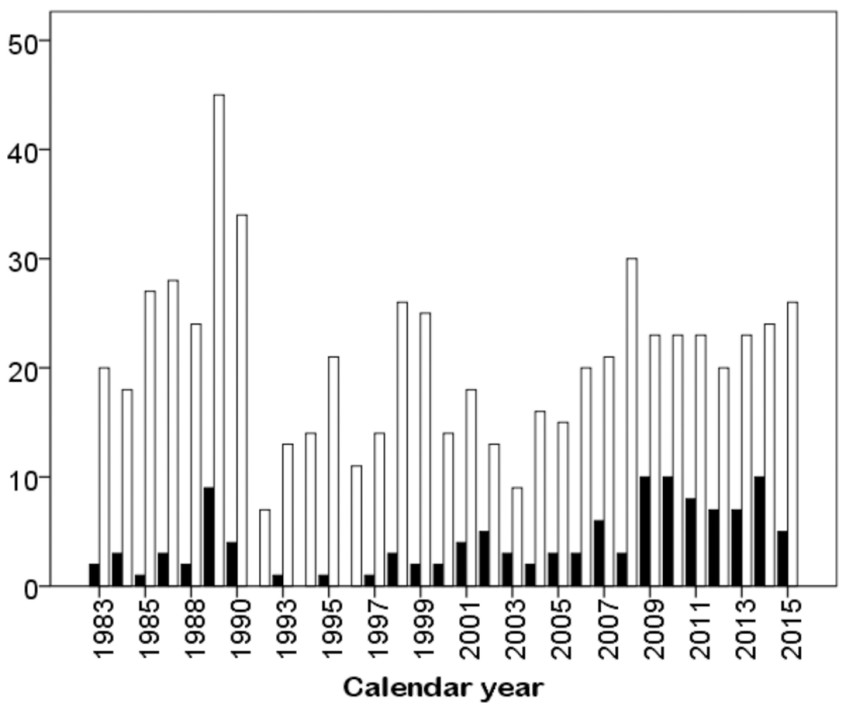

**Figure 2 Participation of women and men in the Hawaii Ultraman from 1983 to 2015.** Women are depicted in dark and men in bright bars.

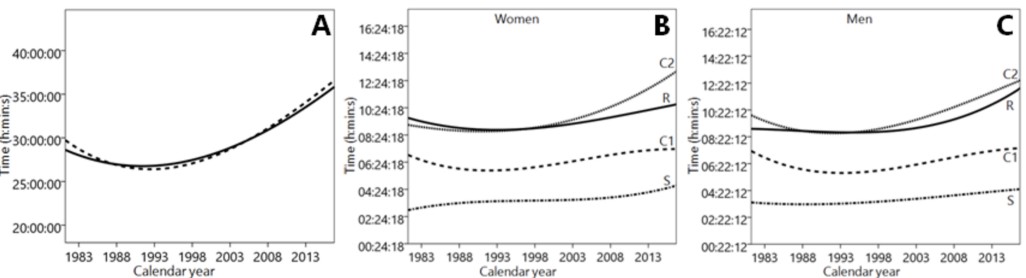

**Figure 3 Variation of female and male participants' performance in the Hawaii Ultraman from 1983 to 2015.** Total race time for women (solid line) and men (dashed line) (left), Race time in swimming (S), cycling 1 (C1), cycling 2 (C2) and running (R) for women (middle) and men (right).

There was a main effect of calendar year on overall race and all race events' times ($p < 0.001$) (Table 2, Fig. 3). A sex × calendar year interaction was observed only in cycling 2 ($p = 0.013$), in which the increase in race time was more pronounced in women than in men. Also, there was a main effect of calendar year on age ($p < 0.001$), in which the age of participants increased during the last years (Table 3 and Fig. 4).

## DISCUSSION

This study investigated pacing trends across stages in 'Ultraman Hawaii' held between 1983 and 2015 with the hypothesis that fast starters would slow down (i.e., increase race

**Table 2  Coefficients (C) and standard errors (SE) from multi-variate regression models for the performance by calendar year in women and men.**

|  |  | C | SE | p |
|---|---|---|---|---|
| **Swimming** | Sex (=female) | 9885.21 | 48918.07 | 0.840 |
|  | Calendar year | 121.80 | 9.26 | <0.001 |
|  | Interaction sex × calendar year | −5.06 | 24.42 | 0.836 |
| **Cycling 1** | Sex (=female) | −58501.28 | 69593.16 | 0.401 |
|  | Calendar year | 148.77 | 13.18 | <0.001 |
|  | Interaction sex × calendar year | 29.40 | 34.74 | 0.398 |
| **Cycling 2** | Sex (=female) | −156860.04 | 62856.90 | 0.013 |
|  | Calendar year | 344.63 | 11.90 | <0.001 |
|  | Interaction sex × calendar year | 78.40 | 31.38 | 0.013 |
| **Running** | Sex (=female) | 118761.88 | 112339.14 | 0.291 |
|  | Calendar year | 233.10 | 21.27 | <0.001 |
|  | Interaction sex × calendar year | −59.33 | 56.08 | 0.290 |
| **Total** | Sex (=female) | −86714.23 | 210194.69 | 0.680 |
|  | Calendar year | 848.30 | 39.80 | <0.001 |
|  | Interaction sex × calendar year | 43.40 | 104.92 | 0.679 |

**Table 3  Coefficients (C) and standard errors (SE) from multi-variate regression models for the age of participants by calendar year in women and men.**

|  |  | C | SE | p |
|---|---|---|---|---|
| **Age** | Sex (=female) | −195.41 | 152.82 | 0.201 |
|  | Calendar year | 0.29 | 0.03 | <0.001 |
|  | Interaction sex × calendar year | 0.10 | 0.08 | 0.203 |

time) compared to slow starters who would improve (i.e., decrease race time). We found, as expected, differences between performance groups and sexes.

The most important finding was that the best women adopted a different pacing strategy during the three days compared to the best men. In details, the fastest overall women decreased split times from swimming to cycling 1 (i.e., day 1) and maintained performance thereafter (i.e., day 2 and day 3), whereas the fastest overall men decreased split times from swimming to cycling 2 (i.e., day 1 and day 2) but increased time in running (i.e., day 3). Expressed in other words, the best women improved performance in the beginning of the race (i.e., day 1), whereas the best men improved performance in both day 1 and day 2, but were slower in the end of the race (i.e., day 3). Potential explanations for these differences could be sex differences in anthropometry, race strategy, separate performance in the three split disciplines and the increase in participation in women in the last years. Therefore, the main sex difference in pacing of the fastest athletes was highlighted in day 2, when men decreased race time, whereas women maintained a steady pacing. This pattern in day 2 might explain the sex differences in day 3, when men increased race time and women maintained steady pacing. These sex differences indicated that performance in a split might affect following performance.

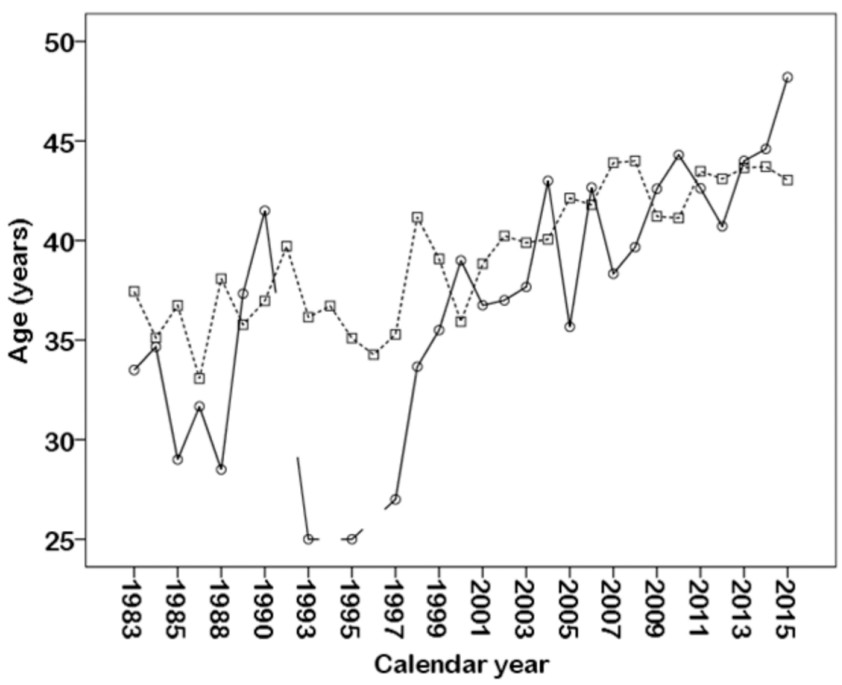

**Figure 4** Variation of female and male participants' age in the Hawaii Ultraman from 1983 to 2015. Women are depicted by solid and men by dashed line.

The findings of the present study were in agreement with the existing literature on sex differences in pacing. For instance, differences in pacing between women and men have been observed in triathlon (*Le Meur et al., 2011*; *Vleck et al., 2008*), but also in other sports disciplines such as running. *Deaner & Lowen (2016)* investigated 3,948 performances in 5-km cross country running and showed that women slowed significantly more than men. These authors argued that the sex difference in pacing partly reflected a sex difference in some aspect of decision making, such as over-confidence, risk perception, or willingness to tolerate discomfort. In running, difference in pacing might vary by distance. In marathon running, however, men are more likely to slow down (i.e., increase split times) than women (*Deaner et al., 2014*). A more recent study, however, found that women, older and faster marathoners are better pacers than men, younger and slower marathoners, respectively (*March et al., 2011*). These differences in pacing in runners are most likely explained by differences in performance level (*Lima-Silva et al., 2010*).

A further explanation for the differences in pacing between the best women and men could be the differences in split and overall performances. Considering performance in split and overall race time, men were faster than women in cycling (i.e., day 1 and day 2) and overall race time, but not in swimming (i.e., day 1) and running (i.e., day 3). We found that the fastest men improved performance in cycling 1 and cycling 2, but impaired on day 3 in running. On the contrary, the fastest women improved performance in cycling 1 and maintained it in cycling 2 and running. A potential explanation could be that women saved energy during the cycling for the running split. When men would have completely exhausted got of the bike to run the next day a double marathon, they would not have been
able to improve performance. Although women and men suffer an energy deficit during an Ironman triathlon (*Kimber et al., 2002*), female Ironman triathlons suffer no decrease in body mass (*Knechtle et al., 2010b*) compared to male Ironman triathletes (*Knechtle et al., 2010a*). Male Ironman triathletes suffer a decrease in skeletal muscle mass most probably due to depletion of glycogen stores of the lower limb (*Mueller et al., 2013*).

Further explanations for the differences in pacing between women and men could be differences in anthropometry, training and previous experience (*Knechtle et al., 2010d*). Considering predictor variables for Ironman triathlon performance, percent body fat was related to race performance in male athletes and to average weekly training in female athletes. Personal best time in an Ironman triathlon was associated with total race time for both male and female athletes (*Knechtle et al., 2010c*). The greater height and associated longer limbs of the men might also provide a biomechanical advantage. However, anthropometric characteristics such as body height and limb length were not investigated in Ironman triathletes (*Knechtle et al., 2010c*; *Knechtle et al., 2010d*). Personal best marathon time is a strong predictor for Ironman triathlon race in both women and men (*Knechtle et al., 2015c*). Also in male marathoners, body fat and running speed during training was a strong predictive variable, not body height or limb length (*Barandun et al., 2012*). For female marathoners, however, circumference of the calf and running speed during training were the most important predictive variables (*Schmid et al., 2012*).

Differences in metabolic predominance between men and women might also explain these findings. For example, women tend to use fat metabolism to a higher degree than men (*Blaak, 2001*). Since fat is such a huge energy source, women may be better at preserving energy and maintaining similar pacing with these ultra-races. Men have a greater glycolytic capacity (*Lundsgaard & Kiens, 2014*) which means they have greater anaerobic and high intensity exercise capacities but of course greater metabolite accumulation leading to greater fatigue (*Bogdanis, 2012*).

A further important finding was that the age of the athletes increased across years and their performance impaired. This is in contrast to findings for 'Ironman Hawaii' where the age of annual top ten female and male triathletes in the 'Ironman Hawaii' increased between 1983 and 2012 while their performances improved (*Gallmann et al., 2014*). The most likely explanation for these disparate findings is the fact that in the analysis for 'Ironman Hawaii' the annual ten fastest were considered whereas in the present analysis all women and men were included in the analysis. Moreover, a small different sex trend in performance was remarkable during the last years, where race time in cycling 2 increased more in women than in men. The increased overall time during the last years was attributed mostly to increased cycling 2 times in women and to increased cycling 2 and running time in men (Fig. 4). To counteract this calendar year trend, this finding implied that coaches should focus on the improvement of cycling 2 in women and on both cycling 2 and running in men.

A limitation of this analysis is the fact that specific characteristics for this race such as alteration in body composition (*Baur et al., 2016*) and cardiac and pulmonary responses during the race (*Seedhouse, Walsh & Blaber, 2006*) were not considered. In addition, since the distance each event is much longer in "Ultraman Hawaii" than other Ironman triathlon events, caution is required in order to generalize the findings of the present study to other

triathlon formats. The strength of the present study was that it included all participants to all races of "Ultraman Hawaii."

## CONCLUSIONS

In 'Ultraman Hawaii,' women pace differently than men where the fastest women decreased performance on day 1 and could then maintain on day 2 and 3 whereas the fastest men impaired performance on day 1 and 2 but improved on day 3. For athletes and coaches, women and men should follow sex-tailored training and race tactics in such events.

### Funding
The authors received no funding for this work.

### Competing Interests
The authors declare there are no competing interests.

### Author Contributions
- Beat Knechtle conceived and designed the experiments, wrote the paper, reviewed drafts of the paper.
- Pantelis T. Nikolaidis conceived and designed the experiments, performed the experiments, analyzed the data, wrote the paper, prepared figures and/or tables, reviewed drafts of the paper.

### Data Availability
The raw data has been supplied as Supplementary File and can also be found at http://ultramanworld.com/.

### Supplemental Information
Supplemental information for this article can be found online at http://dx.doi.org/10.7717/peerj.2509#supplemental-information.

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
