# Peer review of "Sex differences in pacing during ‘Ultraman Hawaii’"

_PeerJ, doi:10.7717/peerj.2509_

## Round 0.1 · original submission · Minor Revisions

Dear Dr. Knechtle,

I have considered the two sets of review comments. Regarding the comments of Reviewer 2, in discussion with the Editorial Office we do believe that the article is 'in scope' for the journal.

Therefore, please respond to all the review comments appropriately. Given Reviewer 1's comments about scope, you may want to improve the manuscript to provide more analysis of course.

Best Regards,

Duane

·

Basic reporting

Fine

Experimental design

Fine

Validity of the findings

Fine

Additional comments

Sex differences in pacing during Ultraman Hawaii’

Abstract:
Background: There is no background in the background section, only an objective. Can the authors provide a sentence or two why this is important perhaps based on previous work or lack thereof.

Methods: The following sentence sounds to me like a result not a description of methodology: “Different trends among events (i.e. swimming, cycling 1, cycling 2 and running) were observed by performance group and by sex.”

Key words are supposed to be words not included in the title. Pacing is both in the title and the key words.

Introduction

I realize that this experiment is not a basic physiological examination of pacing but I am surprised that some papers from Noakes and St-Clair group on the physiological rationale for pacing were not mentioned. I would expect at least 1 sentence maybe two applying their physiological explanation for pacing (i.e. avoidance of catastrophic failure of the entire system, potential reserve of energy for the end of the race etc.)

Statistical Analysis

Line 171: For the ease of readers who are not up to date statisticians, perhaps a one-line definition of a z score might be helpful.

Results

Line 206-207: Authors state: “A sex x calendar year interaction was observed only in cycling 2 (p=0.013). Also, there was a main effect of calendar year on age (p<0.001) (Table 3, Figure 4).” OK something happened but you do not inform the readers of the implication. For instance when you reported a sex x calendar year interaction, you informed the readers that this meant there were more women participating in later years. I do not know the implications of the aforementioned findings.

Discussion
Authors state: “This study intended to investigate …”. It seems to me that you DID investigate …, rather than just having the intention to investigate.

Line 245: Authors state; “could show”. I think Deaner and Lowen showed these findings. Delete could and make show past tense.

Line 250: Authors state: “older, women, and faster are better pacers than younger, men, and slower marathoners”. The syntax is confusing. Is the first part of the sentence referring to faster women or just faster runners of either sex. Similarly, is the second part of the sentence referring to slower men or all slow marathoners?

Lines 253-266: Authors should include a description of the differences in metabolic predominance for men and women. For example, women tend to use fat metabolism to a higher degree than men. Since fat is such a huge energy source, women may be better at preserving energy and maintaining similar pacing with these ultra races. Men have greater glycolytic capacities which of course means they have greater anaerobic and high intensity exercise capacities but of course greater metabolite accumulation leading to greater fatigue.

Line 268: Authors mention anthropometry but only provide a brief mention of body fat. Would not the greater height and associated longer limbs of the men also provide a biomechanical advantage?

Figure 2 legend: Which bars represent women and which bars represent men?
Figure 4 legend: Which lines represent women and which lines represent men?

·

Basic reporting

No comments

Experimental design

No comments

Validity of the findings

No comments

Additional comments

The aim of the manuscript intends to examine the effect of age, sex and calendar year on triathlon performance. Although I recognize the tremendous work behind the manuscript and the effort of running complex statistical plans, no real research question has been addressed. The manuscript resembles more of a technical report to be submitted to sport agencies or sport journals. Besides, the authors refer to race pace, a parameter that they did not record. They use, instead, the time provided on websites that does not inform about the shape of pacing strategies used by the competitors. Therefore, the discussion around the pacing strategies is quite speculative and no mechanisms could be discussed to highlights new scientific insights.

---

## Round 0.2 · accepted · Accept

Dear Beat,

I have completed the review process of your manuscript and now feel that it is acceptable for publication in PeerJ.

Nice work and best of luck in the future.

Duane